# Chemical reaction-mediated covalent localization of bacteria

Huilong Luo[1], Yanmei Chen[1], Xiao Kuang[1], Xinyue Wang[1], Fengmin Yang[1], Zhenping Cao[1], Lu Wang[1], Sisi Lin[1], Feng Wu [1] & Jinyao Liu [1] ✉

Methods capable of manipulating bacterial colonization are of great significance for modulating host-microbiota relationships. Here, we describe a strategy of in-situ chemical reaction-mediated covalent localization of bacteria. Through a simple one-step imidoester reaction, primary amino groups on bacterial surface can be converted to free thiols under cytocompatible conditions. Surface thiolation is applicable to modify diverse strains and the number of introduced thiols per bacterium can be easily tuned by varying feed ratios. These chemically reactive bacteria are able to spontaneously bond with mucous layer by catalyst-free thiol-disulfide exchange between mucin-associated disulfides and newly converted thiols on bacterial surface and show thiolation level-dependent attachment. Bacteria optimized with $9.3 \times 10^7$ thiols per cell achieve 170-fold higher attachment in mucin-enriched jejunum, a challenging location for gut microbiota to colonize. As a proof-of-concept application for microbiota transplantation, covalent bonding-assisted localization of an oral probiotic in the jejunum generates an improved remission of jejunal mucositis. Our findings demonstrate that transforming bacteria with a reactive surface provides an approach to chemically control bacterial localization, which is highly desirable for developing next-generation bacterial living bioagents.

Microbiota-host association, which is mainly related with the behavior of bacterial colonization, plays essential roles in human health[1–3]. Sufficient colonization of beneficial and symbiotic bacteria is critical to prevent and treat diseases via retaining a balanced microbiota structure to maintain the metabolic and immune homeostasis of the host[4–7]. Naturally, microbial colonization majorly depends on the interaction between bacterial surface and various in vivo biological interfaces[8–10]. For example, bacteria evolved to express multiple surface adhesins to promote interfacial adhesion, which facilitates extracellular muco-polysaccharide synthesis and biofilm formation to potentiate colonization[11–13]. Therefore, methods capable of manipulating interfacial interactions between bacteria and the surroundings are of great significance to control over microbial colonization.

Expression of specific membrane ligands or receptors by synthetic bioengineering or introduction of adhesive functional groups on the surface through physicochemical approaches has been extensively explored to improve the interaction of bacteria with host extracellular matrix proteins, such as fibrinogen, collagen, and others[14–17]. For instance, *Escherichia coli* has been genetically encoded with a controllable expression of surface capsular polysaccharides to increase accumulation in target tissues by intervening the interaction with immune cells[18]. Genetic editing of biofilm structures has been exploited to tune the adaptability and responsiveness of *Bacillus subtilis* to dynamically interact with the environments[19,20]. On the other hand, we, along with other groups, have employed physical encapsulation or surface conjugation of bacteria to decrease the affinity with unfriendly

[1]Shanghai Key Laboratory for Nucleic Acid Chemistry and Nanomedicine, Institute of Molecular Medicine, State Key Laboratory of Oncogenes and Related Genes, Shanghai Cancer Institute, Renji Hospital, School of Medicine, Shanghai Jiao Tong University, 200127 Shanghai, China. ✉e-mail: jyliu@sjtu.edu.cn

environmental substances[21–26], while increasing their in vivo reservation at the locations of interest with the help of hydrogen bonding[27–30], electrostatic[31,32], hydrophobic[33,34], and π-π conjugation interactions[35]. However, to the best of our knowledge, previous methods are solely based on noncovalent bonding between bacteria and their surrounding interfaces, inevitably leading to inadequate interactions. Alternative approaches, particularly with the capability of strengthening interfacial interactions to improve the colonization of beneficial and symbiotic bacteria, are hence highly desirable.

Mucin is an important category of large extracellular glycosylated proteins that are main organic components of mucus layer[36]. The oligosaccharide chains consisting of 5–15 monomers exhibit moderate branching and are attached to the protein core by forming O-glycosidic bonds with the hydroxyl groups of serine and threonines and arranged in a bottle brush configuration[37] (Supplementary Fig. 1). Mucin has cysteine-rich domains in N and C terminals that mediate chain extension by end-to-end disulfide linkage of mucin monomers. Cysteine-rich regions are actively abundant and also reported as internal domains that contribute to disulfide side links between intermediate cysteine thiols[38].

Here, we present a concept of manipulating bacterial localization by chemically adjusting bacteria-host interfacial interactions and demonstrate its application to develop bacterial therapeutics, particularly with ability to increase accumulation on the mucin-abundant tissue surface. Primary amino residues abundant on the surface of diverse bacteria can be transformed to free thiols via a simple one-step imidoester reaction under cytocompatible conditions (Fig. 1a). Surface thiolation produces chemically reactive living bacteria that are able to spontaneously link with poly(disulfide)s-enriched mucous layer by forming new disulfide bonds through dynamic thiol-disulfide exchange reaction (Fig. 1b). Due to the ability to form covalent bonds, surface-thiolated bacteria show advantages to target and colonize tissues characterized with thick mucus. In contrast to unmodified species, reactive bacteria with an optimized surface thiol number reach up to 170-fold higher attachment in mucin-abundant jejunum, which represents a challenging location for intestinal microbes to colonize. As a proof-of-concept application example of microbiota transplantation, we convert therapeutic *Escherichia coli* Nissel 1917 (EcN) with a thiolated surface and find that oral delivery of bioorthogonal EcN generates a notably promoted remission of inflammation associated with jejunal mucositis. Given the characteristics of covalent bonding-assisted localization, we anticipate that endowing bacteria with a reactive surface could provide a robust approach to prepare various functional living bioagents.

## Results and discussion
### Design and characterization of surface-thiolated bacteria
We hypothesized that exogenous free thiols could exchange with disulfide groups existing in mucin to form new disulfide bonds. As a proof-of-concept study, EcN strain was selected as the model bacterium given its ability to regulate host microbiota homeostasis and work together with other probiotics to modulate enzyme fabrication, local immunity, and digestion[39,40]. Surface thiolation of EcN was carried out by a simple one-step chemical reaction with 2-iminothiolane (Traut's reagent), a water-soluble cyclic thiomidoester. After mixing in phosphate-buffered saline (PBS, pH 7.4) at room temperature, 2-iminothiolane readily reacted with primary amino groups from N-terminal and lysine residues in peptides/proteins presenting on bacterial surface to introduce free thiol groups.

To ensure negligible influences on bacterial viability and proliferation, the reaction time was first optimized for surface thiolation. The cytotoxicity of 2-iminothiolane on EcN was assessed by bacterial

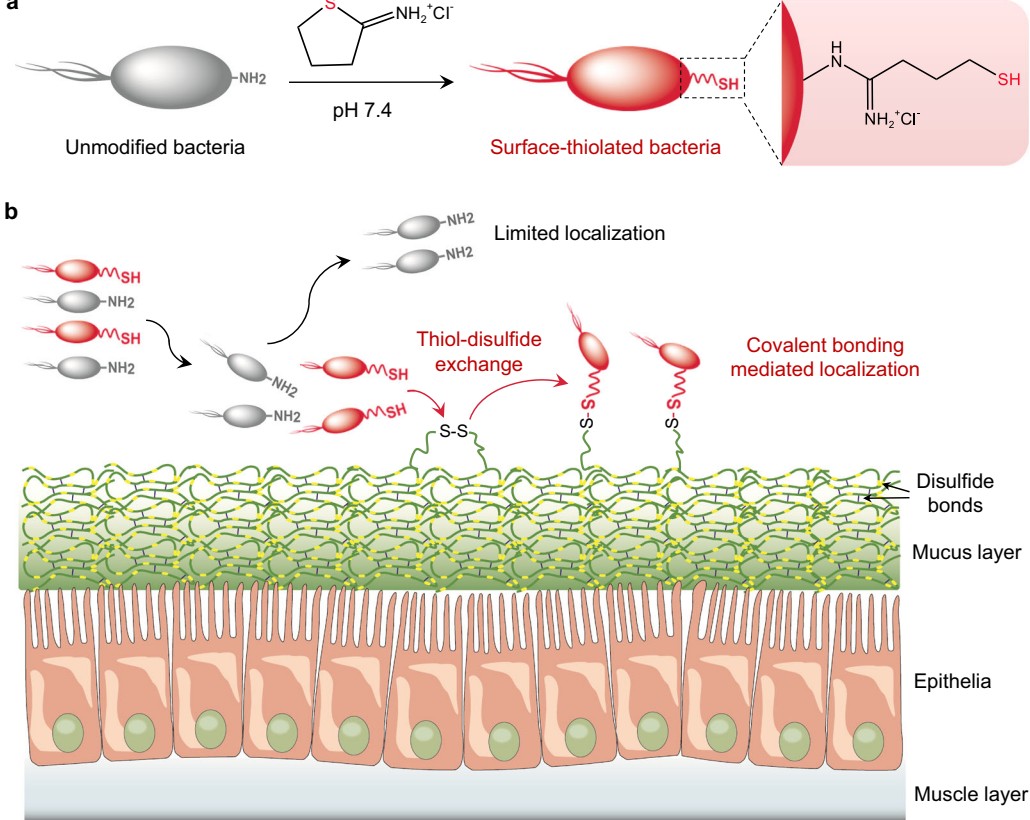

**Fig. 1 | Schematic illustration of chemical reaction-mediated covalent localization of bacteria. a** Preparation of surface-thiolated bacteria by a simple one-step imidoester reaction under cytocompatible conditions. **b** Bonding of chemically reactive surface-thiolated bacteria with poly(disulfide)s-abundant mucin locating at various tissue interfaces by catalyst-free dynamic thiol-disulfide exchange reaction.

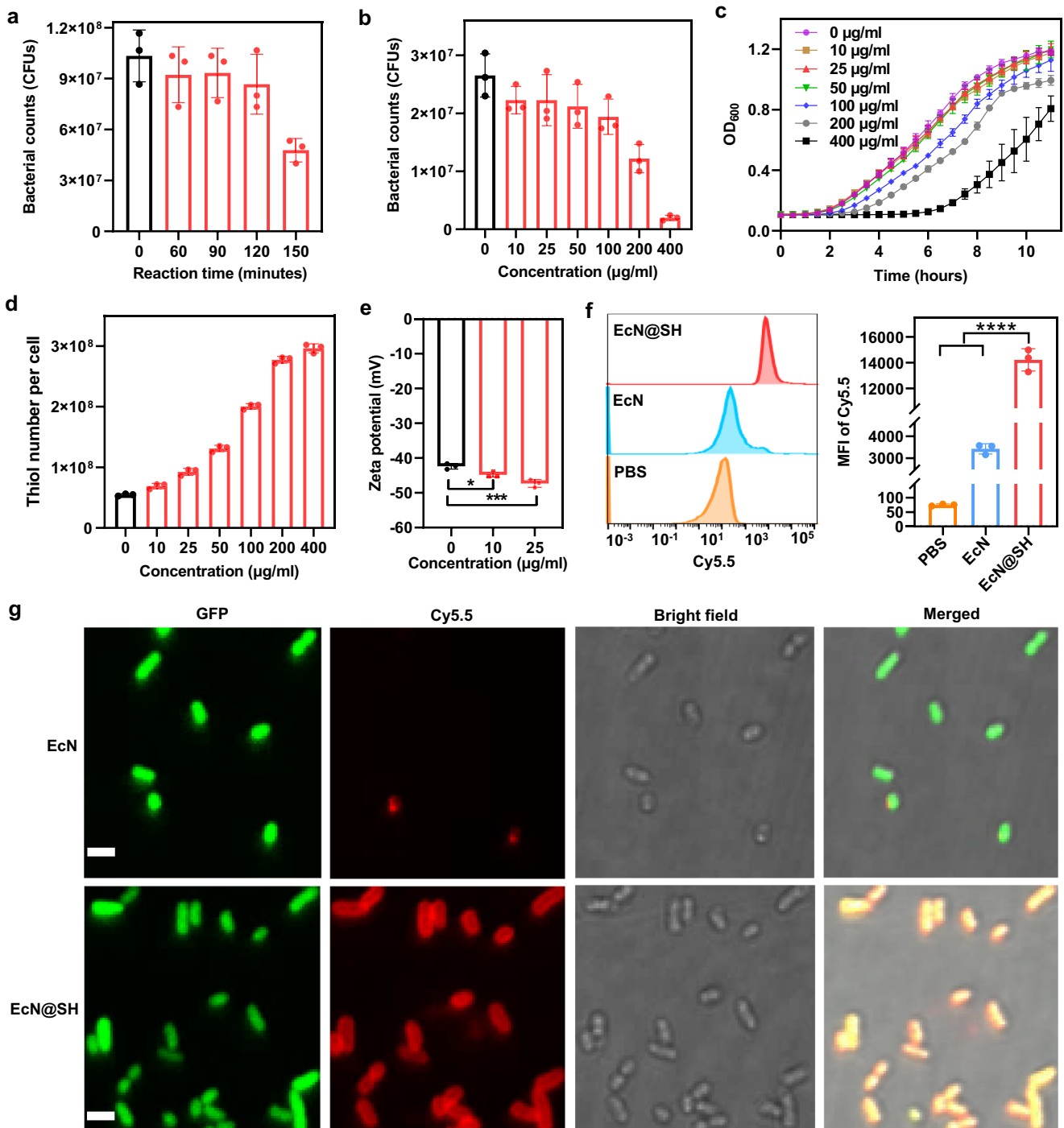

**Fig. 2 | Preparation, characterization, and reactivity of surface-thiolated bacteria. a** Survival of EcN after reaction with 2-iminothiolane for varied time intervals, including 60, 90, 120, and 150 min. **b** Survival, **c** growth curves, and **d** surface thiol numbers of EcN after reaction with different concentrations of 2-iminothiolane ranging from 0 to 400 µg/ml for 90 min. **e** Zeta potentials of EcN after thiolation with different concentrations of 2-iminothiolane. **f** Flow cytometry histograms and MFI values of native EcN and EcN@SH post reaction with Cy5.5-maleimide for 90 min. **g** Typical LSCM images of GFP-expressed unmodified EcN and EcN@SH after reaction with Cy5.5-maleimide. Scale bar: 2 µm. Data are presented as mean values ± SD ($n = 3$, from independent experiments). Significance was assessed using one-way ANOVA analysis followed by Fisher's LSD multiple comparisons, giving $P$ values, *$P < 0.05$, ***$P < 0.001$, ****$P < 0.0001$. Source data and the exact $P$ values are provided in the Source Data file.

counting on selective lysogeny broth (LB) agar plate after reaction with 2-iminothiolane for different time intervals. As shown in Fig. 2a, the viability of EcN was decreased along with reaction time, with a significant reduction observed after prolonging to 150 min. The concentration of 2-iminothiolane ranging from 0 to 400 µg/ml was further screened under a reaction time of 90 min and the trend of viability

variation plotted in Fig. 2b and Supplementary Fig. 2 showed a dose-dependent cytotoxicity. It was noted that limited effects on bacterial viability were observed at a concentration of less than 100 µg/ml. To examine whether surface thiolation affected the growth of EcN, modified EcN (EcN@SH) were cultured with 100% LB and growth curves were recorded over 11 h by monitoring the values of optical

density at 600 nm (OD$_{600}$), illustrating insignificant influence at a concentration lower than 50 µg/ml (Fig. 2c). The total proteins extracted from EcN and EcN@SH were analyzed by Coomassie staining. The overall bacterial protein composition remained unaffected and no additive product of factors was observed after thiolation (Supplementary Fig. 3). Tryptophan metabolism was also chosen to examine the impact of thiolation on the metabolic activity of thiolated bacteria. As numerous bacterial species can metabolize tryptophan into specific metabolites, such as indole and its derivatives[41], the indole test was applied to evaluate the ability of thiolated bacteria to degrade tryptophan by tryptophanase. As claimed in Supplementary Fig. 4a, thiolated EcN demonstrated similar positive results to native bacteria, which produced a ring of purple color of indole in the upper ether layer after supplementing Kovac's reagent. The concentration of the generated indole was quantitatively analyzed by high-performance liquid chromatography. Expectedly, no markable difference was observed in the generation of indole between native and thiolated bacteria, validating that thiolation had negligible impact on the tryptophan metabolism of bacteria (Supplementary Fig. 4b).

We further determined the thiolation level of bacteria using 5,5-dithio-bis(2-nitrobenzoic acid) (DTNB, Ellman's reagent). It was worth mentioning that DTNB exhibited favorable cytocompatibility and bacterial viability remained near unchanged after incubation with DTNB under experimental conditions (Supplementary Fig. 5). The amount of thiol moiety was calculated from a standard curve obtained from solutions with serially diluted concentrations of L-cysteine hydrochloride hydrate (Supplementary Fig. 6). As confirmed in Fig. 2d, even at a low applied 2-iminothiolane concentration of 10 µg/ml, the number of thiols on a single bacterium was increased greatly by 26.5%, with newly added thiols as high as $1.4 \times 10^7$ per cell. The level of thiolation was dose-dependently increased, giving a 50% increment at a 2-iminothiolane concentration of 88.3 µg/ml (Supplementary Fig. 7). With increasing concentration to 25 and 50 µg/ml that exhibited ignorable impacts on bacterial viability and growth, thiol numbers on EcN@SH were largely increased by 1.7- and 2.4-fold in contrast to native EcN, with new thiols upgraded to $3.8 \times 10^7$ and $7.6 \times 10^7$ per cell, respectively. Therefore, to balance thiolation level and bacterial viability, a modest condition of 25 µg/ml 2-iminothiolane and 90 min incubation at room temperature in PBS was adopted for the following experiments unless otherwise indicated. In addition, flow cytometry was applied to determine the level of amine groups on bacteria using cyanine5-N-hydroxysuccinimide (Cy5-NHS). As plotted in Supplementary Fig. 8, thiolated bacteria exhibited similar amine level to native EcN, which might be ascribed to the reactivity of secondary amines converted from the primary amines. Dynamic light scattering (DLS) was performed to measure the changes in size and surface charge after modification, revealing no apparent alteration in particle size and a slight decrease of 2 to 4 mV in zeta potential depending on thiolation levels (Fig. 2e and Supplementary Fig. 9). The decrement in surface potential was in accordance with the negative charge of S$^-$ derived from attached thiols on bacterial surface[42]. Overall, the above results suggested that the use of 2-iminothiolane was able to efficiently convert primary amino groups on bacterial surface to thiols under cytocompatible conditions.

### Reactivity of introduced thiols and versatility of surface thiolation

The reactivity and location of anchored thiols were analyzed using flow cytometry and laser scanning confocal microscopy (LSCM), in which cyanine5.5-maleimide (Cy5.5-maleimide) was applied as a selective click-labeling reagent. As claimed in Fig. 2f (left panel), after reaction with Cy5.5-maleimide for 90 min at room temperature, native EcN displayed a slight fluorescent shift, which resulted from the reaction of inherent thiols with the maleimide group of fluorescence dye. In contrast, EcN treated with 25 µg/ml 2-iminothiolane emerged a near

complete shift post reaction with Cy5.5-maleimide, evidencing a ~4.1-fold increment in the mean fluorescence intensity (MFI) of Cy5.5 (Fig. 2f, right panel). More importantly, LSCM images clearly demonstrated the formation of a Cy5.5-maleimide labeled thiol layer on the surface of individual EcN@SH expressing GFP (Fig. 2g and Supplementary Fig. 10), with a signal-to-background ratio increased by 12.3-fold in comparison to native EcN (Supplementary Fig. 11). This result clarified that thiolation mediated by 2-iminothiolane indeed occurred mainly on bacterial surface. It was worth mentioning that although free intrinsic thiols were detected for bacteria, their reactivity towards Cy5.5-maleimide was relatively lower than that of chemically transformed thiols based on flow cytometry and LSCM measurements. This founding could be explained by that the majority of native bacterial thiols were embedded in complex three-dimensional protein structures[43], while extension with an 8-A ring spacer endowed chemically transformed thiols with higher flexibility for further reaction[44].

We further validated the versatility and cytocompatibility of this approach to modify other bacteria. Several bacterial strains including both Gram-negative *Akkermansia muciniphila* (AKK) and *Salmonella typhimurium* (STM) as well as Gram-positive *Enterococcus faecalis* (EF), *Bacillus megaterium de Bary* (BMB), and *Bacillus cereus* (BC) were thiolated under the optimized condition. Comparable to that of EcN@SH, zeta potentials of AKK, STM, EF, BMB, and BC were decreased separately from $-32.0 \pm 0.3$, $-26.2 \pm 1.6$, $-26.0 \pm 0.4$, $-36.0 \pm 0.4$, and $-32.8 \pm 0.7$ mV to $-33.3 \pm 0.8$, $-28.3 \pm 0.8$, $-27.3 \pm 1.1$, $-42.0 \pm 0.4$, and $-40.2 \pm 0.3$ mV post thiolation (Supplementary Fig. 12). After reaction with Cy5.5-maleimide, MFI values of thiolated AKK, STM, EF, BMB, and BC were largely increased by 3.1-, 4.1-, 35.9-, 39.6-, and 31.1-fold, respectively (Fig. 3a, b, e, f, i, j, m, n, q, r). Interestingly, relative low heterogeneity of fluorescent signal for different native bacteria were demonstrated after reaction with Cy5.5-maleimide, with MFI ranging from 2000 to 8000, which was ascribed to varied numbers of intrinsic thiols (Fig. 3b, f, j, n, r). The presence and location of thiols was also visualized by LSCM imaging, showing a uniform Cy5.5-marked layer around each thiolated bacterium (Fig. 3c, g, k, o, s), while negligible signal on native bacteria (Supplementary Figs. 13 and 14). It was found that thiolation level of Gram-positive bacteria was around 8- to 20-fold higher than those of Gram-negative bacteria (Fig. 3b, f, j, n, r). This might be explained by the existence of more primary amines on the surface of Gram-positive bacterial strains, which has been reported previously[45]. Similarly, no negative influence was observed for the viability and growth of these thiolated bacteria, further supporting the high compatibility of this thiolation strategy (Fig. 3d, h, l, p, t). Taken together, cytocompatible chemical thiolation was applicable for diverse bacterial species, which could be endowed with numerous reactive thiols on the surface for further reaction.

### Reaction between surface-thiolated bacteria and mucin

Having confirmed the reactivity of thiols on the surface of thiolated bacteria, we next studied catalyst-free reaction between modified EcN and mucin via dynamic thiol-disulfide exchange (Fig. 4a). The change of surface-free thiol groups on modified bacteria over time was investigated in PBS using flow cytometry. As displayed in Supplementary Fig. 15, half of the newly-formed thiols disappeared with incubation time increasing to 2 h. The level of thiols decreased much more slowly after 2 h of incubation and returned to the baseline of native EcN with time prolonging to 12 h. These results suggested a relatively long reservation of the new-formed thiols on EcN@SH surface for thiol-disulfide exchange reaction. To facilitate detection, Cy5.5-labeled mucin (Cy5.5_mucin) was prepared by esterification with Cy5.5-NHS for 3 h at 37 °C and the product was purified by ultrafiltration. After direct incubation of $3 \times 10^8$ colony forming units (CFUs) of thiolated EcN with 0.03 mg/ml of Cy5.5_mucin for 1 h, the MFI of the individual bacterium was remarkably increased in contrast to native bacteria (Fig. 4b). A similar increment was observed with mucin

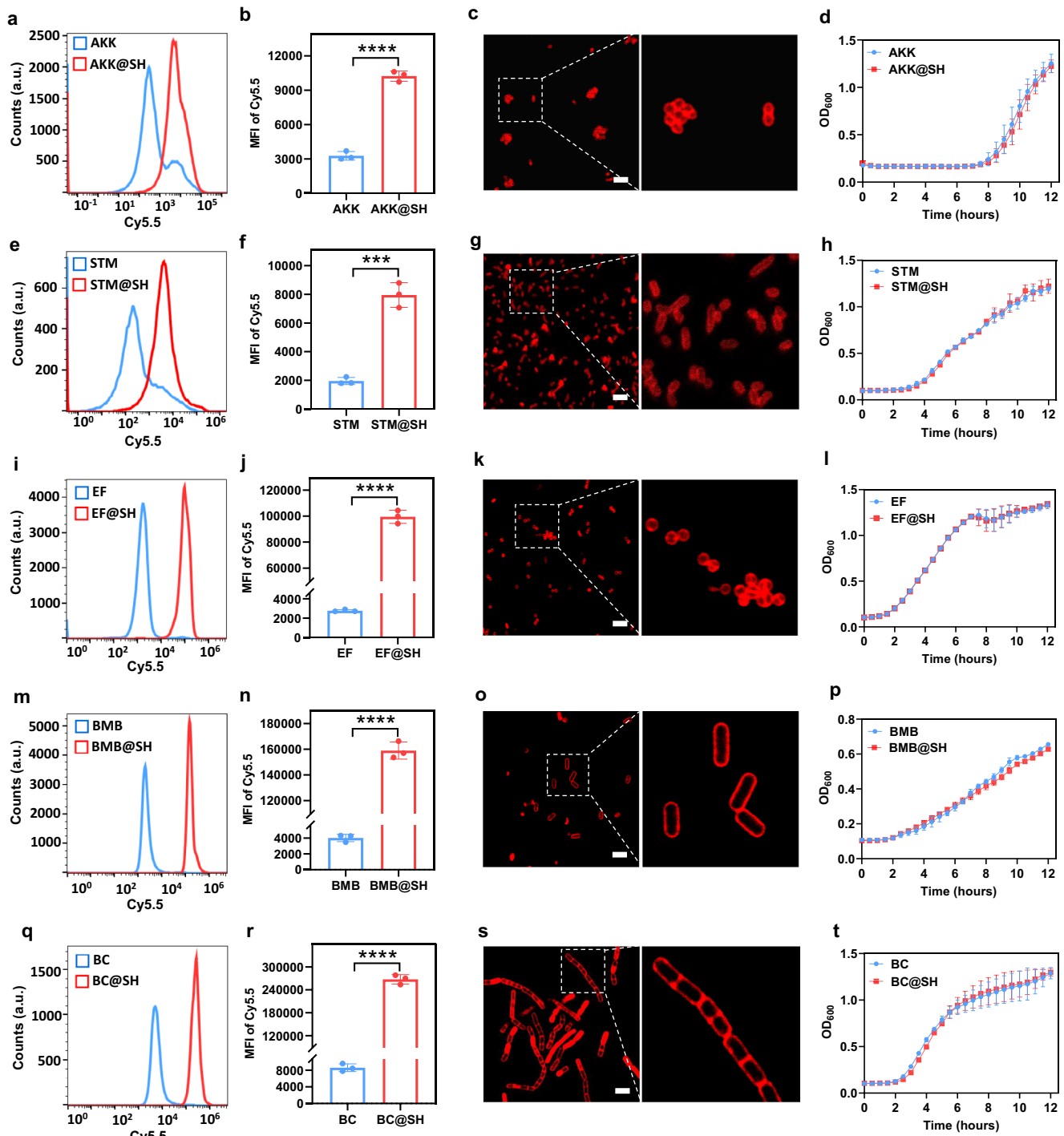

**Fig. 3 | Surface thiolation of diverse bacterial species. a, e, i, m, q** Flow cytometry histograms of thiolated (**a**) AKK, (**e**) STM, (**i**) EF, (**m**) BMB, and (**q**) BC post reaction with Cy5.5-maleimide for 90 min. Unmodified bacteria were used as controls. **b, f, j, n, r** MFI values of unmodified bacteria and corresponding thiolated (**b**) AKK, (**f**) STM, (**j**) EF, (**n**) BMB, and (**r**) BC after reaction with Cy5.5-maleimide detected by flow cytometric analysis. **c, g, k, o, s** Typical LSCM images of the thiol layers on thiolated (**c**) AKK, (**g**) STM, (**k**) EF, (**o**) BMB, and (**s**) BC. Scale bar: 5 μm.

**d, h, l, p, t** Growth curves of thiolated (**d**) AKK, (**h**) STM, (**l**) EF, (**p**) BMB, and (**t**) BC. Native bacteria were used as controls. The a.u. indicates arbitrary units. Data are presented as mean values ± SD ($n = 2$–4, from independent experiments). Significance was assessed using Student's $t$ test (two-tailed), giving $P$ values, ***$P < 0.001$, ****$P < 0.0001$. Source data and the exact $P$ values are provided in the Source Data file.

concentration increasing to 0.1 mg/ml (Supplementary Fig. 16), implying the conjugation of Cy5.5-mucin onto thiolated bacteria. To investigate the influence of thiolation level on mucin attachment, we altered the density of thiol groups on bacterial surface and found that the MFI of EcN was increased significantly with the number of thiols increasing from $5.5 \times 10^7$ to $9.3 \times 10^7$ per cell (Fig. 4c, d). Thiolation

level-dependent attachment indicated that the interaction between thiolated bacteria and mucin could be tuned easily by varying the number of surface thiol groups. To verify the involvement of thiols in bioconjugation, we utilized thiol-sensitive Cy5.5-maleimide to record the change of thiol number after reaction with mucin. Expectedly, thiol quantity represented by Cy5.5 fluorescence intensity was reduced

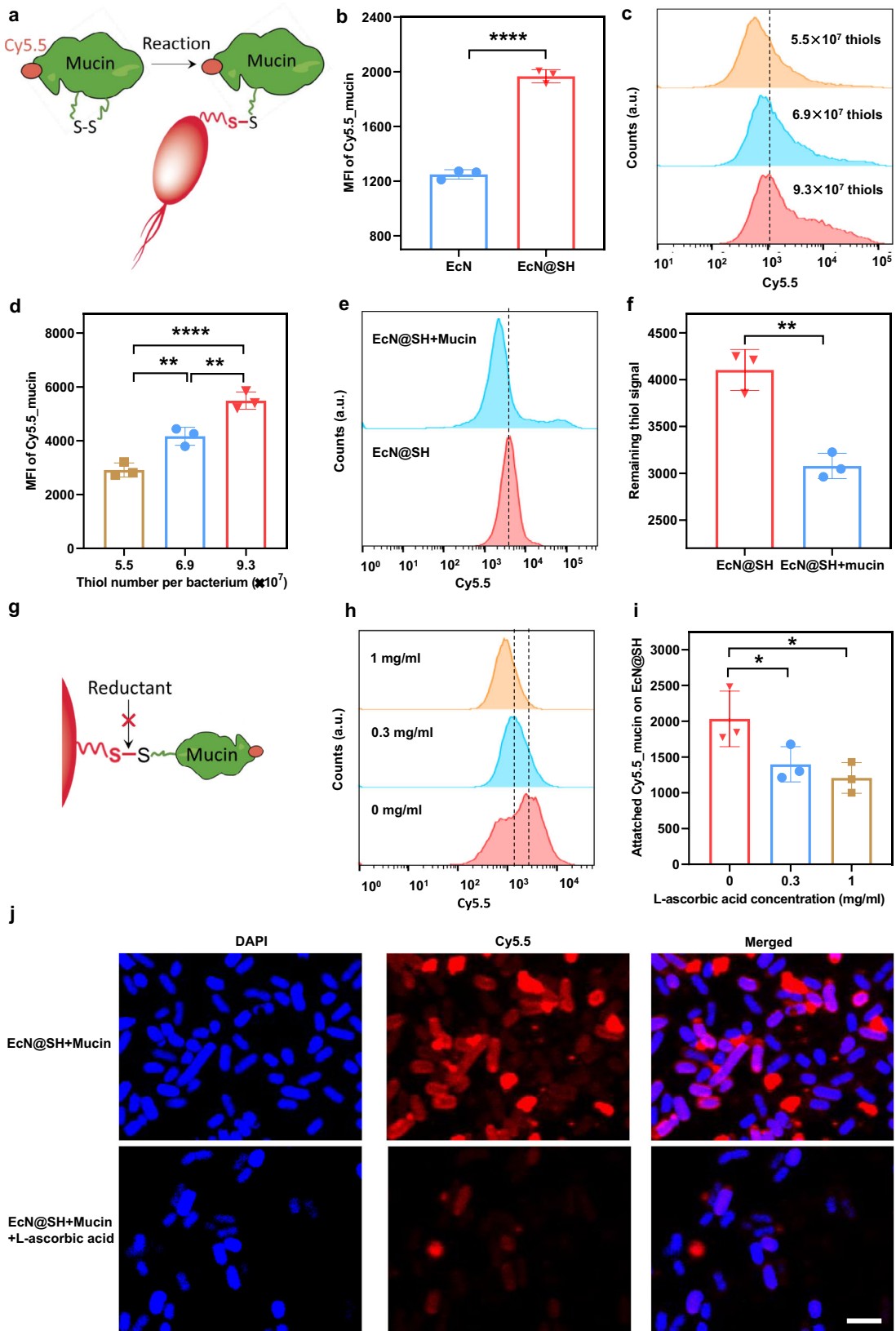

significantly for thiolated EcN after incubation with mucin, proving the consumption of free thiols during reaction (Fig. 4e, f). Moreover, DLS was conducted to detect the changes in size and surface potential after mucin attachment, showing increased particle size and a slight decrease of ~1–2 mV in zeta potential in comparison to native EcN (Supplementary Figs. 17 and 18). To validate the formation of disulfide bonds between thiolated bacteria and mucin, L-ascorbic acid was added into the mixture prior to reaction given that thiol-disulfide exchange could be dampened by reductive agents[46] (Fig. 4g). At a L-ascorbic acid concentration of 0.3 mg/ml, thiolated EcN after reaction with Cy5.5_mucin presented a clear decrement in MFI, which was further reduced with concentration increasing to 1 mg/ml (Fig. 4h, i).

**Fig. 4 | Reaction between surface-thiolated bacteria and mucin. a** Schematic illustration of catalyst-free reaction between thiolated bacteria and mucin via dynamic thiol-disulfide exchange. **b** MFI values of unmodified EcN and EcN@SH with $8.5 \times 10^7$ thiols per cell post reaction with 0.03 mg/ml of Cy5.5_mucin for 1 h. **c** Flow cytometry histograms and **d** MFI values of Cy5.5_mucin-attached EcN@SH with different surface thiol densities after reaction with 0.1 mg/ml of Cy5.5_mucin for 1 h. **e** Flow cytometry histograms and **f** MFI values of mucin-attached EcN@SH after reaction with Cy5.5-maleimide. **g** Schematic illustration of blocked thiol-disulfide exchange by reductants. **h** Flow cytometry histograms and **i** MFI values of

EcN@SH after reaction with 0.03 mg/ml of Cy5.5_mucin in the presence of 0, 0.3, or 1 mg/ml of L-ascorbic acid. **j** Representative LSCM images of EcN@SH after reaction with Cy5.5_mucin in the presence or absence of 1 mg/ml of L-ascorbic acid. Scale bar: 3 μm. The a.u. indicates arbitrary units. Data are presented as mean values ± SD ($n = 3$, from independent experiments). Significance was assessed using Student's $t$ test (two-tailed) or one-way ANOVA analysis followed by Fisher's LSD multiple comparisons, giving $P$ values, $^*P < 0.05$, $^{**}P < 0.01$, $^{****}P < 0.0001$. Source data and the exact $P$ values are provided in the Source Data file.

Namely, the conjugation of Cy5_mucin with thiolated bacteria was affected by L-ascorbic acid, which was ascribed to the blockade of thiol-disulfide exchange reaction rendered by reductants. Meanwhile, similar results were obtained by LSCM imaging, showing that the presence of fluorescently labeled Cy5_mucin on thiolated EcN was dramatically inhibited after adding L-ascorbic acid (Fig. 4j). These results were in good agreement with a previous report, where cellular uptake mediated by thiol-disulfide exchange could be suppressed by a reductant[47]. Moreover, dithiothreitol (DTT), which can reduce disulfide bonds, was applied to cleave the newly-formed disulfides between EcN@SH and mucin. As displayed in Supplementary Fig. 19, the attached mucin on EcN@SH could be completely cleaved by 0.1 mg/ml of DTT, verifying covalent bond-mediated attachment of mucin on EcN@SH via thiol-disulfide exchange. In addition to reductive agents, selenocystamine (SeCA), which contains diselenium and acts as a mimetic substrate competing with disulfides to exchange with thiols[48], was used to inhibit the reaction between EcN@SH and mucin. As plotted in Supplementary Fig. 20, 0.3 mg/ml of SeCA greatly blocked the conjugation of mucin on EcN@SH, further suggesting the occurrence of a thiol-disulfide exchange reaction. Briefly, these data confirmed the reactivity of thiolated bacteria to link with cysteine-enriched mucin by forming disulfide bonds through thiol-disulfide exchange.

## Covalent localization of thiolated bacteria on the mucus layer

Jejunal microbiota has been found to be closely associated with diabetes, obesity, and mucositis induced by alcohol, radiation, and chemotherapy[49,50]. However, the jejunum refers to a challenging location for transplanted microbiota to colonize due to the existence of unfavorable microenvironments and physical barriers[51–53], as reflected by its low level of $10^3$–$10^5$ CFUs/ml bacterial abundance[54], which is in stark contrast to $10^{11}$–$10^{12}$ CFUs/ml in the colon[55,56]. Thus, overcoming the defense against bacterial attachment in the jejunum remains of high interest for enhanced microbial transplantation. Inspired by the coverage of an abundant disulfide-rich mucus layer on the inner lining of the jejunum[57], we speculated that covalent bonding between thiolated bacteria and mucin might be able to promote attachment. To verify this hypothesis, sampled mouse jejunum was everted and exposed to thiolated EcN expressing mCherry for 1 h and fluorescence intensity of bacteria attaching onto jejunal mucosa was measured by in vivo imaging system (IVIS) (Fig. 5a). Compared to native EcN, exposure to EcN@SH exhibited a stronger fluorescence signal, indicating enhanced adhesion of thiolated bacteria onto mucosal layer (Fig. 5b). Quantitative analysis showed a significant increase compared to native EcN (Fig. 5c), supporting thiol-disulfide exchange-assisted attachment of EcN. The sectioned jejunal segments were further homogenized for bacterial counting, testifying a 6.9-times increment in the number of EcN attached to the mucosal layer of jejunum (Fig. 5d). Encouraged by these improvements, we also evaluated the validity of covalent bonding-mediated attachment in an ex vivo porcine model. Swine, which is arguably one of the most proper models of human organ systems, particularly the skin and mucosal system, has provided numerous translational advantages as a preclinical model[58]. The adhesion of EcN@SH to jejunal mucosa was assessed using corresponding tissues of Bama miniature pigs with

body weight around 20 kg. Related to native EcN, a markedly higher reservation of fluorescence signal was detected for thiolated EcN (Fig. 5e, f), which claimed a 3.5-times increment in bacterial attachment by plate counting (Fig. 5g). Covalent bonding of thiolated bacteria with mucin proposed a simple yet robust chemical approach to reinforce bacterial localization on mucosal layer across species.

We then explored the in vivo attachment and localization of EcN@SH following oral ingestion in mice. Jejunal tissue and its mucus were separately collected at 1, 12, and 24 h after administration of $1.0 \times 10^8$ CFUs of EcN@SH. Both PBS and unmodified EcN were dosed as controls. Representatively, jejunal mucus collected at 1 h post oral gavage was observed by LSCM. To distinguish dosed EcN from endogenous bacteria, EcN with an anti-kanamycin marker were transfected with a plasmid containing an mCherry gene. As expected, no EcN was visualized for mucus sampled from mice administered with PBS (Fig. 5h). Despite fluorescently emitted EcN could be observed in the sample of unmodified EcN-gavaged mice, corresponding bacterial number was low. Differently, a large number of red-fluorescent EcN were emerged under LSCM visualization of jejunal mucus harvested from mice dosed with EcN@SH. The counts of EcN localizing at jejunal tissue and its mucus were quantified by plate numeration at different time intervals (Fig. 5i). As plotted in Fig. 5j, the counts of EcN@SH retained in jejunal mucus were far exceeded than those of unmodified EcN at all the indicated time points. At 1 h post oral gavage, the number of EcN@SH colonizing jejunal mucus reached as high as $1.7 \times 10^7$ CFUs/g, which was 171.3 times higher than that of mice administered with native EcN. Even with time post administration extending up to 24 h, increment in the localization of thiolated EcN in jejunal mucus was remained as high as 44.0-times in comparison to that of native EcN-dosed mice. Comparable increment trend was acquired from the sampled jejunal tissues and 40.6-, 15.9-, and 9.6-fold increments were separately achieved at 1, 4, and 24 h after gavage (Fig. 5k and Supplementary Fig. 21), further certifying thiol-disulfide exchange-based localization of EcN in mouse jejunum after oral administration. Given that thiol-disulfide exchange could be blocked by reductive agents, L-ascorbic acid was orally administered along with thiolated EcN. As expected, the localization of EcN@SH in jejunal mucus was largely increased compared to that of native EcN. However, the increased localization was dramatically decreased by 88-fold after co-administering with 1 mg/ml of L-ascorbic acid (Supplementary Fig. 22). L-ascorbic acid-triggered decrement in EcN@SH localization in jejunal mucus validated the covalent bonding of thiolated bacteria with mucin in vivo. In addition, intestinal length measurement and hematoxylin and eosin (H&E) staining ensured that no detectable damages and inflammatory responses was induced after transplantation of thiolated bacteria (Supplementary Figs. 23 and 24).

## Application of thiolated EcN for mucositis remission

Lastly, we turned our attention to evaluate whether thiolated bacteria could be exploited to remit inflammation associated with jejunal mucositis, which is caused by the side effects of chemotherapy, radiotherapy, and anti-metabolite agents[59]. Taking advantage of clinically used EcN in treating inflammatory bowel diseases[30], we speculated that chemical reaction-enabled attachment and localization of transplanted EcN@SH could promote the remission of jejunal

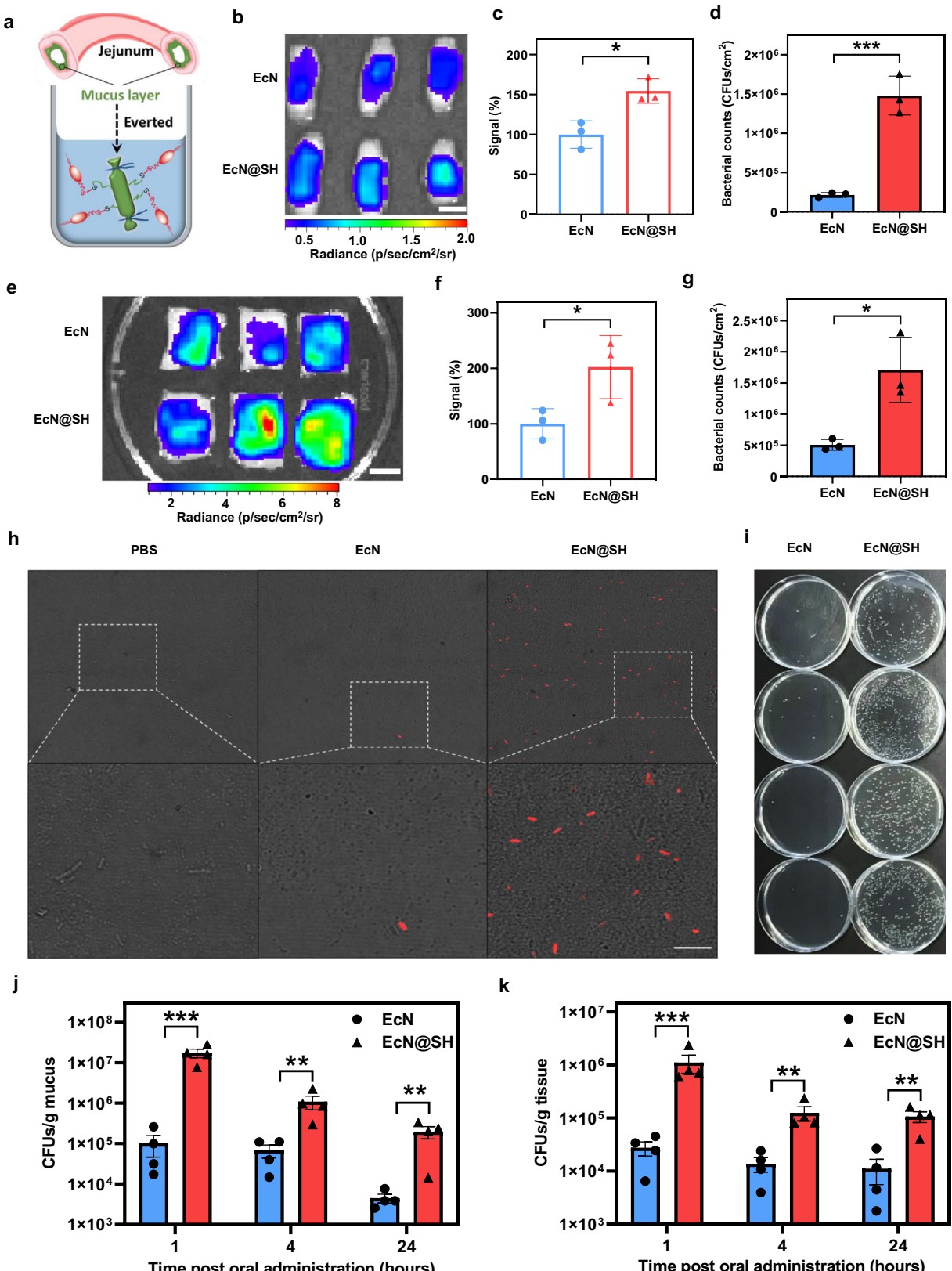

mucositis (Fig. 6a). To demonstrate this, the potential of thiolated EcN was assessed in a 5-fluorouracil-induced (5-FU-induced) murine model of jejunal mucositis[60], which is characterized by shortened lengths of the intestine and villus, cytokine abnormalities, and histopathological damages of intestinal tissues[61,62] (Supplementary Fig. 25). After daily administration with $1 \times 10^8$ CFUs of EcN@SH per mouse for 5 days,

mice were euthanized for sample collection. Healthy mice were used as a positive control, while PBS and equivalent unmodified EcN were utilized as negative controls. As given in Fig. 6b, dosing with EcN alone generated negligible beneficial effects, as reflected by insignificant increment in intestinal length in comparison to those of PBS mice. Gavage of EcN@SH achieved comparable intestinal length to that of

**Fig. 5 | Covalent localization of thiolated bacteria on mucus layer. a** Schematic illustration of experimental design for examining reaction efficiency between thiolated bacteria and jejunum mucus layer. **b** IVIS images and **c** corresponding fluorescence intensities of everted murine jejunal segments after cultivation with equivalent unmodified EcN or EcN@SH expressing mCherry. Scale bar: 0.5 cm. **d** Numbers of bacteria attached to murine jejunal mucus layer by plate counting. **e** IVIS images and **f** corresponding fluorescence intensities of porcine jejunal segments after incubation with equivalent native EcN or EcN@SH expressing mCherry. Scale bar: 3 cm. **g** Numbers of bacteria attached to porcine jejunal mucus layer by plate counting. **h** Representative LSCM images of mouse mucus layer collected at 1 h after oral administration of PBS, unmodified EcN, or EcN@SH. mCherry-expressed EcN were used. Scale bar: 10 μm. **i** Typical digital images of agar plates containing EcN collected from mouse jejunal mucus at 1 h after oral delivery of unmodified EcN or EcN@SH. Counts of EcN and EcN@SH collected from **j** jejunal mucus and **k** jejunal tissue at 1, 4, and 24 h after oral gavage, respectively. **c, d, f, g** Data are presented as mean values ± SD (*n* = 3, from independent biological samples). **j, k** Data are presented as mean values ± SD (*n* = 4 mice). Significance was assessed using Student's *t* test (two-tailed), giving *P* values, *$P < 0.05$, **$P < 0.01$, ***$P < 0.001$. Source data and the exact *P* values are provided in the Source Data file.

positive control, which was significantly prolonged than both of PBS and EcN groups. The levels of inflammatory cytokines, including tumor necrosis factor-alfa (TNF-α) and interleukin-6 (IL-6) in EcN@SH-dosed mice, were reduced apparently in contrast to those of PBS and EcN controls (Fig. 6c and Supplementary Fig. 26). In addition, the inflammation of jejunal tissue was assessed by myeloperoxidase (MPO) staining, showing significantly less MPO positive cells in jejunal lesion after EcN@SH treatment (Supplementary Fig. 27). Furthermore, EcN@SH remitted the inflammatory responses, vacuolization, and edema in the jejunum more efficiently according to H&E staining of jejunal sections (Fig. 6d). Quantitative histopathology analysis clarified that EcN@SH-administered mice exhibited the lowest disorders in villus length and crypt depth among all treated groups (Fig. 6e–g). Immunofluorescence staining of jejunal tissue appeared decreased and discontinued expressions of ZO-1 and occludin in both negative control groups, which were greatly reversed by EcN@SH (Fig. 6h). Elevated levels of tight junction proteins disclosed a protective effect of EcN@SH in restoring the integrity of the intestinal barrier. Similar to healthy mice, recoveries in crypt depth as well as the lengths of intestine and villus, relief of inflammation, alleviated histopathological damages, and strengthened gut barrier by EcN@SH affirmed the values of thiolated bacteria for covalent bonding-mediated microbial therapy.

In summary, we have described the use of a chemical strategy to manipulate the interaction of bacteria with environmental interfaces by forming covalent bonds. This approach refers to a few characteristics: (1) a proposal of chemically reactive living cellular agents; (2) a conception of covalent bonding-mediated interaction between bacteria and surroundings; (3) an example of in vivo in-situ chemical reaction-enabled bacterial localization; and (4) the demonstration of successful transplantation of microbiota to the jejunum. The introduction of covalent bonding is more robust and durable compared to prior modifications, such as chemical coating and physical encapsulation, which result in noncovalent inadequate interactions and can be destroyed or eliminated after a few rounds of bacterial replication. Despite chemical modification is relatively temporary in contrast to synthetic bioengineering, potential safety issues of gene contamination associated with genetic engineering can be avoided. In short, our work opens a window for chemically modulating microbiota-host interaction and provides a versatile platform to prepare various innovative bacterial agents for broad biomedical applications.

## Methods

All the experiments were complied with international guidelines and animal experiments were performed under the guidelines evaluated and approved by the ethics committee of the Institutional Animal Care and Use Committee of Shanghai Jiao Tong University.

### Materials

The involved chemicals and biologicals were summarized as below: 2-iminothiolane (Traut's reagent, molecular weight 137.63, Thermo Scientific™), mucin from porcine stomach (Type III, Sigma-Aldrich), 5,5-dithio-bis(2-nitrobenzoic acid) (DTNB, Ellman's reagent, Thermo Scientific™), L-ascorbic acid (BBI Life Sciences), 5-fluorouracil (5-FU, Adamas), lysogeny broth (LB, BBI Life Sciences), cyanine5.5-maleimide (Cy5.5-maleimide, Aladdin), and cyanine5.5-*N*-hydroxysuccinimide (Cy5.5-NHS, Aladdin). Cy5.5-labeled mucin (Cy5.5_mucin) was prepared by esterification with Cy5.5-NHS for 3 h at 37 °C and the residual free Cy5.5-NHS and other impurities were purified by ultrafiltration. LB liquid medium was fabricated with 25 g of LB broth in 1 L of deionized (DI) water and used after autoclavation. LB agar plates were prepared on bacterial dishes with 8 ml of LB agar solution (containing 25 g of LB broth and 15 g of agar in 1 L of DI water). All other materials were provided by domestic suppliers and applied as received.

### Bacterial strains and plasmids

*Escherichia coli* Nissle 1917 (EcN), *Akkermansia muciniphila* (AKK), *Salmonella typhimurium* (STM), *Enterococcus faecalis* (EF), *Bacillus megaterium* de Bary (BMB), and *Bacillus cereus* (BC) were purchased from China general microbiological culture collection center (GMCC, China). Plasmids pBBR1MCS2-Tac-GFP and pBBR1MCS2-Tac-mCherry were obtained from domestic suppliers and applied to express intracellular GFP and mCherry in EcN.

### Conversion of primary amine groups to thiols on the bacterial surface

An EcN colony was taken from an LB agar plate and cultured in LB liquid medium overnight (200 rpm, 37 °C). Then, EcN cells were washed with DI water twice and resuspended in 987.5 μL ice-cold phosphate buffer (PBS, pH 7.4). Afterward, 12.5 μL of 2-iminothiolane solution (2 mg/ml) was added to the suspension, followed by 90 min of incubation at room temperature, leading to an efficient thiolation on bacterial surface. Please note: the final reaction concentrations (10, 25, 50, 100, 200, and 400 μg/ml) and reaction time (60, 90, 120, and 150 min) of 2-iminothiolane were applied to optimize reaction conditions. Thiolated EcN (EcN@SH) were further washed with PBS twice to eliminate residual 2-iminothiolane. Prepared EcN@SH and native EcN suspended in PBS were stocked at 4 °C. Bacterial numbers were determined via making serious dilutions of bacterial suspensions, culturing them on selective LB agar plates overnight (200 rpm, 37 °C) and counting the colony forming units (CFUs).

### Determination of the number of thiol groups

Ellman's reagent (DTNB) was used to quantify the amount of thiol groups on modified bacteria. The bacteria were resuspended in PBS with $3 × 10^8$ CFUs/ml and DTNB was added to achieve a final working concentration of 0.1 mM, with the reaction proceeding for 2 h at room temperature. After removing the precipitated bacteria by centrifugation (6000×*g*, 5 min), 200 μL of the supernatant fluid was transferred to a microtitration plate and the absorbance was immediately measured at 412 nm (microplate reader, BioTek, USA). The amount of thiol moieties was calculated from a standard curve obtained from solutions with increasing concentrations of L-cysteine hydrochloride hydrate. Native EcN were applied as a control for the determination of thiol groups. Meanwhile, the numbers of viable EcN and EcN@SH after 0.1 mM Ellman's reagent treatment were enumerated on selective LB plates.

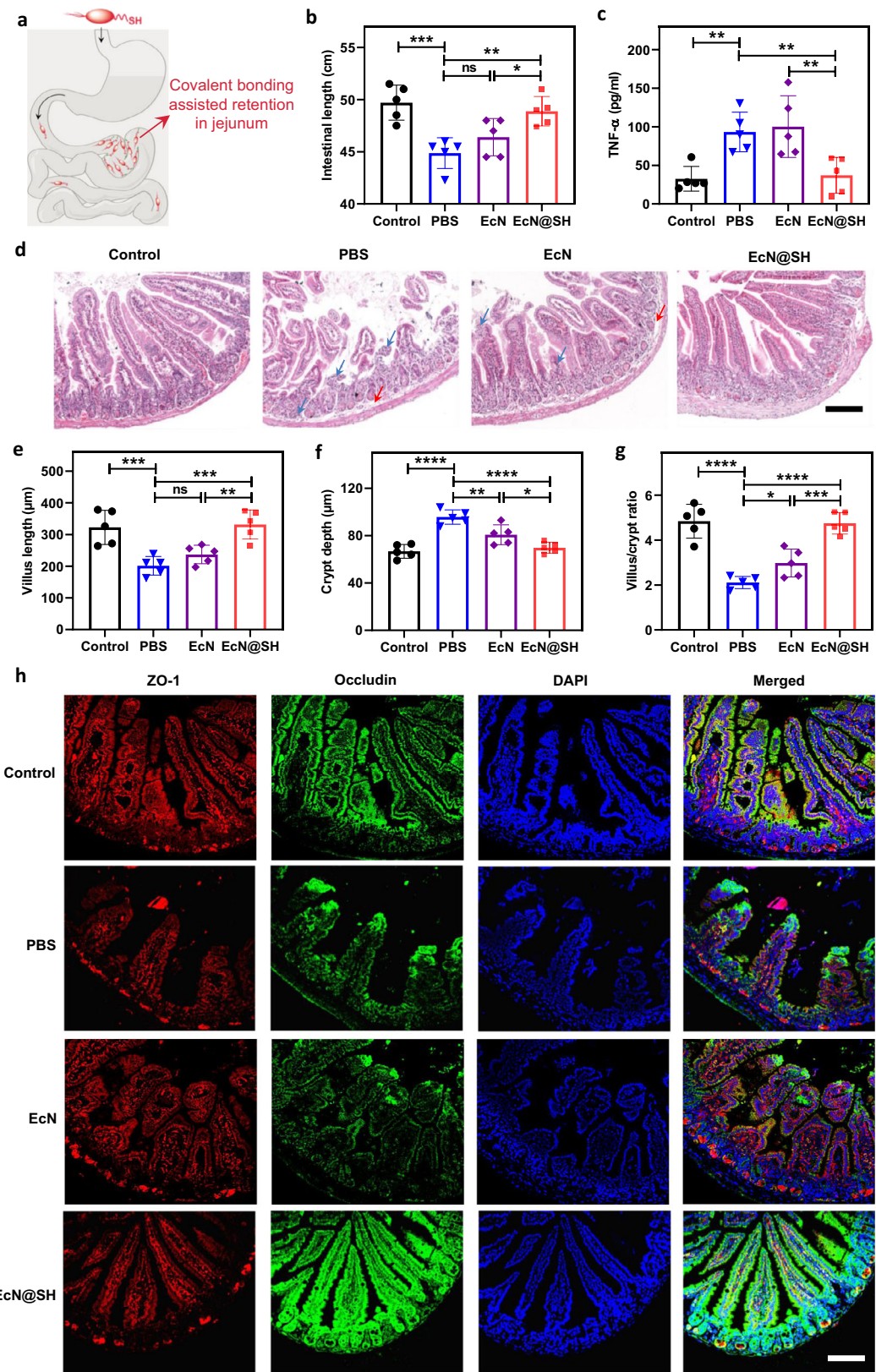

## Characterization of thiolated bacteria

The particle size and zeta potential of thiolated bacteria were determined by dynamic light scattering (DLS) in double-distilled $H_2O$ (ddH$_2$O) at room temperature. Thiolated bacteria were also characterized by laser scanning confocal microscopy (LSCM, Leica TCS SP8, Germany) and flow cytometry (Beckman CytoFlex, USA). Briefly,

prepared thiolated bacteria were mixed with Cy5.5-maleimide solution (0.01 mg/ml) and incubated for 20 min at room temperature allowing for the thiol-maleimide clickable reaction to label the thiol group. Afterward, the reactivity and location of anchored thiols on EcN@SH were subjected to LSCM or flow cytometry analysis. Native bacteria were applied as a control for the above characterization. Flow

**Fig. 6 | Remission of jejunal mucositis by thiolated EcN. a** Schematic illustration of promoted remission of jejunal mucositis by covalent bonding-mediated localization of thiolated EcN. Mice were daily gavaged with PBS, $1 \times 10^8$ CFUs unmodified EcN, or equivalent EcN@SH for 5 days and a dose of 150 mg/kg per mouse of 5-FU was injected intraperitoneally on day 3. Mice were euthanized at day 6 for sample collection. Healthy mice were used as a control. **b** Average intestinal length after treatment. **c** Level of TNF-α in serum measured by commercially available ELISA kits. **d** Typical images of H&E staining of the jejunum. Blue arrows show inflammation and red arrows indicate vacuolization and edema. Scale bar: 100 μm. Quantitative analysis of **e** villus length, **f** crypt depth, and **g** villus/crypt ratio in the jejunum. **h** Immunofluorescence images of ZO-1 and occludin staining of the jejunum. Scale bar: 100 μm. Data are presented as mean values ± SD ($n = 5$ mice). Significance was assessed using one-way ANOVA analysis followed by Fisher's LSD multiple comparisons, giving $P$ values, *$P < 0.05$, **$P < 0.01$, ***$P < 0.001$, ****$P < 0.0001$. ns no significance. Source data and the exact $P$ values are provided in the Source Data file.

cytometry was applied to determine the amine level on bacteria using Cy5-NHS, which was acquired on a CytoFLEX and analyzed by CytExpert (Beckman Coulter, USA) and FlowJo v10 (TreeStar, USA). The total protein of EcN and EcN@SH was analyzed by Coomassie blue staining (P0017F, BeyoBlue™ Super Fast Staining Solution).

### Growth curves of the native and thiolated bacteria

After reaction with 2-iminothiolane, the bacteria were diluted and put in a 96-well plate with $5 \times 10^5$ cells/well in the medium. The plate was incubated at 37 °C with gentle shaking. The absorbance at 600 nm ($OD_{600}$) was recorded for 11 h with each interval of 0.5 h by a microplate reader (BioTek, USA). Native bacteria were applied as a control.

### Metabolic activity of thiolated bacteria

Indole metabolism was used to qualitatively and quantitatively assess the metabolic activity of thiolated bacteria. Equivalent native and thiolated EcN ($2 \times 10^8$ CFUs) were incubated in 3 ml of sterilized peptone medium (0.5% NaCl and 1% peptone in distilled water, pH 7.6) at 37 °C for 48 h. At the settled time point, several drops of ethyl ether were added in peptone medium. Then, two drops of Kovac's reagent were supplemented to qualitatively visualize the generation of indole. The quantity of indole generation was determined by high-performance liquid chromatography (HPLC, Agilent & 1260 Infinity II, USA) with the conditions as below: Diamonsil C18 column (5 μm, 150 × 4.6 mm), column temperature 45 °C, detection wavelength 260 nm, mobile phase (0.1% formic acid in acetonitrile and 0.1% formic acid in water).

### Reaction between surface-thiolated bacteria and mucin in vitro

The catalyst-free reaction between modified EcN and mucin via dynamic thiol-disulfide exchange was testified through in vitro experiments. Native and thiolated EcN were resuspended in ice-cold buffer ($3 \times 10^8$ CFUs/ml, 0.99 ml), 10 μL of 3 mg/ml Cy5.5_mucin dissolved in water was added to achieve a final concentration of 0.03 mg/ml, allowing for reacting at room temperature for 1 h under a 750 RPM/minute shaking. At the indicated time points, the native and thiolated EcN were centrifuged (6000×$g$, 5 min) and washed twice with PBS to completely eliminate the unattached Cy5.5_mucin. Bacteria after reaction with Cy5.5_mucin were sent for flow cytometric analysis. The experiments were conducted under the same procedures with the final concentration of Cy5.5_mucin increasing to 0.1 mg/ml. To investigate the influence of thiolation level on mucin attachment, the density of thiol groups on bacterial surface was altered with the number of thiols increasing from $5.5 \times 10^7$ to $9.3 \times 10^7$ per cell. The bacterial suspension with different thiolation levels was further incubated and reacted with Cy5.5_mucin (0.1 mg/ml) according to the same protocols mentioned above. Meanwhile, to verify the involvement of thiols in the conjugation, thiol-sensitive Cy5.5-maleimide was utilized and analyzed by flow cytometry to record the change of thiol number after reaction with mucin. Lastly, to validate the formation of disulfide bonds between thiolated bacteria and mucin, L-ascorbic acid at varied concentrations (0, 0.3, and 1 mg/ml), dithiothreitol (DTT, 0.1 mg/ml) or selenocystamine (SeCA, 0.3 mg/ml) was added into the mixture prior to the reaction between thiolated EcN and Cy5.5_mucin given that thiol-disulfide exchange could be dampened by these reagents.

The conjugation of Cy5.5_mucin on thiolated EcN was observed by LSCM imaging and also quantified by flow cytometry.

### Ex vivo adhesion of thiolated bacteria toward murine and porcine jejunum

Freshly murine jejunum tissue was divided into ~1.5-cm segments and everted in order to expose the intestinal mucosal layer. The two ends of everted intestine were carefully ligated. The sectioned jejunal segments were randomly divided into two groups ($n = 3$ for each group) and soaked in PBS containing equal numbers of EcN or EcN@SH ($1 \times 10^8$ CFU/ml, expressing mCherry with anti-kanamycin property) for 1 h at a slightly up-down shaking (10 RPM/min). At the indicated time points, the intestinal segments were washed with gently flushed PBS once time and then soaked twice in distilled water in order to completely eliminate unattached bacteria. The intestinal segments were then sent for imaging under in vivo imaging system (IVIS) and counting on selective LB plates post homogenization in PBS. The validity of covalent-mediated attachment in an ex vivo porcine model with the protocols slightly modified due to its large size. The sectioned porcine jejunum was fixed on foam plates using needles with the mucosal layer upside. The prepared EcN or EcN@SH ($3 \times 10^8$ CFU/ml, expressing mCherry with anti-kanamycin property, 20 μL/cm²) were added on the mucosal side and allowed for contacting for 1 h. At the settled time, the tissues were flushed gently using PBS for 10 s. The porcine jejunal segments were sent for IVIS imaging and then homogenized in PBS for further plate counting.

### Covalent localization of thiolated bacteria on jejunal mucus layer in mice

Mice (ICR, female, 6–8 weeks, $n = 4$ mice/group) were fasted for 15 h prior to experiments. EcN were transfected with a plasmid expressing mCherry with an anti-kanamycin marker. The mice were administered with the EcN or EcN@SH dose of $3 \times 10^8$ CFU in 150 μL PBS via oral gavage. The mice administered with PBS were set as a control. At predetermined time points (1, 4, and 24 h), the mice were euthanized and the jejunal tissues were harvested. To evaluate the localization of the bacteria, the jejunum and their respective mucus layers were separately collected. The mucus (10 μL) was spread on glass slides and sent for LSCM imaging. The jejunal tissues and the mucus were further ground and diluted with 1 ml of PBS. The suspensions (40 μL) from both the tissue and the mucus were taken and spread on the selective solid LB plates (100 μg/ml kanamycin). The bacteria were incubated overnight at 37 °C before counting. To validate the formation of disulfide bonds between thiolated bacteria and mucin in vivo, L-ascorbic acid (1 mg/ml) was orally administered along with $3 \times 10^8$ CFU of thiolated EcN in 150 μL PBS. The mice administered with an equivalent amount of unmodified EcN were set as a control. The mice were euthanized at 1 h post oral gavage and jejunal tissues were harvested, homogenized, and enumerated on LB plates.

### 5-FU induced mouse model of jejunal mucositis

Mice (Balb/c, male, 6–8 weeks, $n = 5$ mice/group) were randomly assigned to four groups and orally administered with EcN or EcN@SH ($1 \times 10^8$ CFU/mouse/day) or PBS for 5 days. 5-FU (150 mg/kg mouse) was injected intraperitoneally once on day 3. At day 6, the mice were euthanized for sample collection. Healthy mice were used as a control.

The intestinal length was measured. Serum samples were obtained by centrifugation at $4000 \times g$ for 5 min. The concentrations of TNF-α and IL-6 were detected using commercially available enzyme-linked immunosorbent assay (ELISA) kits (MultiSciences Biotech, China). The jejunal tissues were sampled for blinded histopathology analysis. Jejunal samples were fixed in 4% paraformaldehyde, processed according to standard procedures for paraffin embedding, sectioned at 4 μm, followed by hematoxylin and eosin (H&E) staining and myeloperoxidase (MPO) staining, as well as immunofluorescence staining of ZO-1 and Occludin. The used antibodies included anti-rabbit ZO-1 (1:300; AF5145, Affinity), anti-mouse occludin (1:300; sc-133256, Santa Cruz), goat anti-rabbit IgG (Alexa Fluor 594, 1:500; ab150080, Abcam) and goat anti-mouse IgG (Alexa Fluor 488, 1:500; ab150113, Abcam). Villi and crypts in each intestinal region were measured.

## Statistical analysis

All statistical analysis was evaluated using GraphPad Prism 8. The statistical significance was determined using Student's $t$ test (two-tailed) or one-way ANOVA analysis followed by Fisher's LSD multiple comparisons. The differences between experimental groups and control groups were considered statistically significant at $P < 0.05$. *$P < 0.05$, **$P < 0.01$, ***$P < 0.001$, ****$P < 0.0001$. ns, no significance.

## Reporting summary

Further information on research design is available in the Nature Portfolio Reporting Summary linked to this article.

## Data availability

All data are available within the article or Supplementary information. Source data are provided with this paper.

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

## Acknowledgements

This work was supported by the National Key Research and Develop-ment Program of China (2021YFA0909401), the National Natural Science Foundation of China (21875135, 82204503, 32101218, 22105123, 32201144), and the Innovative Research Team of High-Level Local Universities in Shanghai (SHSMU-ZDCX20210900).

## Author contributions

J.L. supervised the project. J.L. conceived and designed the experiments with H.L. H.L., Y.C., X.K., X.W., F.Y., Z.C., L.W., S.L., and F.W. performed all experiments. All authors analyzed and discussed the data. H.L. and J.L. wrote the paper.

## Competing interests

The authors declare no competing interests.
