## [Peer Review File · Nature Communications]

Chemical reaction-mediated covalent localization of bacteriaREVIEWER COMMENTS

Reviewer #1 (Remarks to the Author):

The paper by Luo et al. described a method to covalently modify the surface of bacteria with thiol groups aimed at increasing the binding of the bacteria to the mucin molecules in the gut. This would aid in benefiting human health such as more efficient delivery of probiotics or microbiota transplants for disease, for example. The study is quite comprehensive with respect to the different types of bacteria employed and some of the methods of characterising both the extent of thiolation and efficiency of bacterial binding to both *in vitro* and *ex vivo* mucin surfaces as well as binding *in vivo* in a mouse model.

The method for thiolation using an imidoester of proteins is not new (Org. Lett. 2014, 16, 5, 1298–1301; ACS Appl. Bio Mater. 2022, 5, 6, 3023–3037) but the authors did optimise the reaction conditions to ensure minimal influence on bacterial viability. Different sub-strains were used e.g. native *E. Coli*, GFP and mCherry *E. coli*. Is there a guarantee that these express the same extra-cellular proteins for thiolation? Despite this, very large increases in bacteria were observed to attach to mucin of explanted jejunum. However, the claim that covalent bonding of thiolated bacteria with mucin occurs in tissue (p.11) is not proved, as little is known about how the bacteria change after thiolation such as product of factors that might facilitate greater attachment to at least the *ex vivo* and *in vivo* systems studied.

Also, how can the authors be sure the thiolation is taking place on the amines of the protein regions of bacteria, given the potential steric hindrance/reaction with the EPS chains? Was the bacterial surface free amine concentration determined? And why are the zeta potential values negative if there are so many free amines available for the thiolation? And why do the zeta values further decrease after thiolation, do S- groups actually exist to account for that or are they SH groups, and how does that influence conjugation? What prevents the bacteria from crosslinking to themselves after thiolation? Was the bacterial particle size unchanged by DLS? And what about DLS of mucin coated bacteria? Very little information is provided on how the zeta potential measurements were performed, e.g. pH, ionic strength can influence the results. There could have also been attempts to characterise the modifications and conjugations in the model system using chemical techniques (e.g. FTIR or XPS, for example)

Mucins are a highly structurally diverse family of glycoproteins and perhaps the structure depicted in Figure 1 and S1, and description on p.5 is too simplistic. Are the disulphide regions of the mucin accessible? Similar to the point above there could be steric hindrance/reaction by/with the sugar chains hindering disulphide formation. How specific really is the thiolation reaction? Finally, what was the purity of the commercially used mucin? The reviewer's experience is that these can be contaminated with surfactants.

A highlight of the paper is the preliminary *in vivo* data in mice that showed significant changes to jejunum physical properties, and slight reductions in inflammation as measured by reductions in expression of by one cytokine (TNF- α), when tested in a 5-fluorouracil-induced murine model of jejunal mucositis. It would have been good to increase the library of cytokine and chemokines tested to gauge more about the inflammatory pathways, and a discussion on why there was a small reduction in inflammation would be useful.

Grammar and spelling needs checking

Overall this is quite a novel study but there are assumptions about the chemical modification on bacteria that need further clarification.

Reviewer #2 (Remarks to the Author):

In this manuscript, Luo et al. described a chemical reaction-mediated approach that facilitates covalent colonization of gut bacteria. The bacterial surface amine was converted into a free thiol in a one-step imidoester reaction. According to the authors, the free thiol group in the modified bacteria will help it spontaneously bond with a mucous layer rich in disulfides and remain attached longer. As a proof of concept, they applied this reaction to *E. coli* nissile (EcN). They demonstrated that thiolation

of its surface increases its attachment time to mucus in the jejunum. Additionally, prolonged attachment ameliorates jejunal mucositis. There have been several studies showing that chemical modifications of bacterial surface structures either enhance or decrease their attachments to mucus (PMID: 11014524 and papers cited in the manuscript). There are several major issues that need to be addressed:

1. The first comment is more conceptual. The definition of bacterial colonization is the 'presence of a microorganism on/in a host, with growth and multiplication of the organism, but without interaction between host and organism'. Based on their description, the free thiol group is added in vitro, and only the modified bacteria are introduced in vivo. Each time the bacteria replicate in the intestine, the surface thiol groups will be diluted. The modification facilitates the attachment of the modified bacteria, but it is unclear whether this modification improves Ecn colonization and facilitates the bacteria's growth and amplification to occupy an intestinal niche more efficiently. It would be more appropriate if the authors revised the manuscript to focus on the attachment/adhesion of Ecn rather than colonization.
2. The second comment is more technical. The chemical 2-iminothiolane they use is a small molecule. Did the authors examine whether this compound can be transported into Ecn cells and interact with cellular free amines? This also applies to the Ellman's reagent they used to quantify the thiol numbers. The cells were treated with this reagent to determine the thiol number. 1M is a quite high concentration. Will the cells get lysed during the treatment? There is a possibility that the total thiols quantified include both surface and intracellular free thiols.
3. The half-life of the modified bacteria in vitro and in vivo. The authors need to assess how long (or how many rounds of replication) it takes for the modified bacteria to lose their surface free thiol groups in vitro. In their in vivo experiments, the authors measured CFU levels at 1, 4, and 24 hours after gavage. As the CFU of modified Ecn decreases, the authors need to determine how long it will take for Ecn to return to its baseline level of attachment.
4. It remains unclear how their method (covalent conjugation) is superior to prior approaches. The purpose of chemically modifying bacterial surfaces is to increase their attachment to epithelium. If the same result can be achieved via other methods (Page 4, lines 2 to 3), then the authors should state clearly and also demonstrate why their method is superior to previous ones. For example, why would noncovalent interactions between bacteria and epithelium result in inadequate interactions? While covalent bonding will not. All these chemical modifications are somewhat 'temporary' and do not occur in vivo. One round of bacterial replication may decrease surface free thiol groups by half. And in their experiments the modified bacterium was still administered daily.

Reviewer #1 (Remarks to the Author):

The paper by Luo et al. described a method to covalently modify the surface of bacteria with thiol groups aimed at increasing the binding of the bacteria to the mucin molecules in the gut. This would aid in benefiting human health such as more efficient delivery of probiotics or microbiota transplants for disease, for example. The study is quite comprehensive with respect to the different types of bacteria employed and some of the methods of characterising both the extent of thiolation and efficiency of bacterial binding to both in vitro and ex vivo mucin surfaces as well as binding in vivo in a mouse model.

Response: We thank the reviewer very much for her/his positive review of our work and providing many useful comments and constructive suggestions. The raised concerns have been addressed appropriately. Please see below for in detailed point-by-point responses.

1. The method for thiolation using an imidoester of proteins is not new (Org. Lett. 2014, 16, 5, 1298–1301; ACS Appl. Bio Mater. 2022, 5, 6, 3023–3037) but the authors did optimise the reaction conditions to ensure minimal influence on bacterial viability. Different sub-strains were used e.g. native E. Coli, GFP and mCherry E. coli. Is there a guarantee that these express that same extra-cellular proteins for thiolation? Despite this, very large increases in bacteria were observed to attach to mucin of explanted jejunum. However, the claim that covalent bonding of thiolated bacteria with mucin occurs in tissue (p.11) is not proved, as little is known about how the bacteria change after thiolation such as product of factors that might facilitate greater attachment to at least the ex and in vivo systems studied.

Response: We thank the reviewer for acknowledging our efforts in optimizing the reaction conditions to ensure minimal influence on bacterial viability.

For the sub-strains, since the plasmids of pBBR1MCS2-Tac-GFP and pBBR1MCS2-Tac-mCherry used in the section of “Bacterial strains and plasmids” in p.17 were applied to express intracellular GFP and mCherry proteins, the extra-cellular protein types and levels would theoretically not be altered. We have addressed this concern by supplementing extra experiments. As shown below (Fig. A), under the same reaction conditions (25 µg/ml 2-iminothiolane and 90 minutes incubation at room temperature), similar thiolation levels were observed across these sub-strains.

To prove the covalent bonding of thiolated bacteria with mucin, we have taken into account the suggestion of the reviewer and conducted extra experiments. Given that thiol-disulfide exchange could be blocked by reductive agents, L-ascorbic acid (1 mg/ml) was orally administered along with thiolated EcN. As expected, EcN@SH localized in jejunal mucus was largely increased than that of mice administered with native EcN. However, the localization was dramatically decreased by 88-fold after co-administering with L-ascorbic acid (Fig. S22, corresponding to Fig. B as shown below). L-ascorbic acid-triggered decrement in EcN@SH localization in jejunal mucus validated the covalent bonding of thiolated bacteria with mucin in vivo. We have also considered the suggestion of the reviewer and two extra experiments were

conducted to explore the changes of bacteria after thiolation. First, to determine whether additional factors were produced after thiolation, the total proteins extracted from EcN and EcN@SH were visualized by Coomassie staining assays. Actually, the overall bacterial protein composition remained unaffected and no additive factors were observed after thiolation (Fig. S3, corresponding to Fig. C as shown below). Second, tryptophan metabolism was chosen to examine the impact of thiolation on the metabolic activity of the thiolated bacteria. As numerous bacterial species can metabolize tryptophan into specific metabolites, such as indole and its derivatives, the indole test was applied to evaluate the ability of thiolated bacteria to degrade tryptophan by tryptophanase. As claimed in Fig. S4a (corresponding below Fig. D, left panel), thiolated EcN demonstrated similar positive results to native bacteria, which produced a ring of purple color of indole in the upper ether layer after supplementing Kovac's reagent. The concentration of the generated indole was also quantitatively analyzed by high-performance liquid chromatography. Expectedly, no markable difference was observed in the production of indole between native and thiolated bacteria, validating that thiolation had negligible impact on the tryptophan metabolism of bacteria (Fig. S4b, corresponding to below Fig. D, right panel). Thus, the above two extra experiments demonstrated that the thiolated bacteria remained un-altered in protein composition and metabolic activity and the enhanced colonization mainly relied on covalent bonding of thiolated bacteria with mucin.

A**B****C****D**
2. Also, how can the authors be sure the thiolation is taking place on the amines of the protein regions of bacteria, given the potential steric hindrance/reaction with the EPS chains? Was the bacterial surface free amine concentration determined? And why are the zeta potential values negative if there are so many free amines available for the thiolation? And why do the zeta values further decrease after thiolation, do S^- groups actually exist to account for that or are they SH groups, and how does that influence conjugation? What prevents the bacteria from crosslinking to themselves after thiolation? Was the bacterial particle size unchanged by DLS? And what about DLS of mucin coated bacteria? Very little information is provided on how the zeta potential measurements were performed, e.g. pH, ionic strength can influence the results. There could have also been attempts to characterise the modifications and conjugations in the model system using chemical techniques (e.g. FTIR or XPS, for example).

Response: We are grateful to the reviewer for highlighting these issues.

The applied thiolation agent, 2-iminothiolane, is a small molecule with a molecular weight of 137.63, which can overcome the resistance from potential steric hindrance/reaction with EPS chains. Actually, the thiolation can also occur on EPS chains, as which contain abundant glycoproteins that present primary amine groups (Nature Reviews Microbiology, 2022, 1, 17).

2-Iminothiolane can specifically react with primary amines to form sulfhydryl groups. In addition to the formation of a thiol group, the primary amine is converted into a secondary amine and a new imine group is introduced by this reaction (Fig. 1a, corresponding to below Fig. A). According to the reviewer's comment, we conducted an extra experiment to determine the bacterial surface free amine concentration using cyanine5-N-hydroxysuccinimide (Cy5-NHS). Flow cytometric analysis showed similar level of amine concentration on thiolated bacteria (Fig. S8, corresponding to below Fig. B), which might be ascribed to the reactivity of the converted secondary amine groups.

The negative zeta potential of the bacterial membrane is majorly attributed to the existences of a large quantity of negatively charged phosphates and carboxylates (Nature Nanotechnology, 2018, 13, 1182). The slight decrease of zeta potential of EcN@SH could be ascribed to the generation of thiol groups. In fact, the groups of SH and S^- always maintain ionization equilibrium in solution, which means that more S^- groups exist on bacterial surface after thiolation (J. Org. Chem. 2008, 73, 12). Meanwhile, as a nucleophile, an activated thiol group attacks one of the two sulfurs in the disulfide bond to form a new disulfide bond (Nature communications, 2021, 12, 163), leading to covalent conjugation of mucin on bacterial surface.

The main factors that prevent bacteria from crosslinking themselves after thiolation could be 1) potential steric hindrance/reaction, 2) the large size and three-dimensional structure of bacteria, which can inhibit their contact possibility and the efficiency of the reaction, and 3) the limited efficiency of SH-SH oxidation without adding catalysts or oxidizers (Nature Reviews Chemistry, 2017, 1, 13).

The measurement of particle size after thiolation was supplemented in the sections of "Design, preparation and characterization of surface-thiolated bacteria" (p.7) and "Reaction between surface-thiolated bacteria and mucin" (p.11). As shown

in Fig. S9, no apparent alteration in particle size was observed after thiolation. We also conducted further experiments to detect the changes in size and surface potential after mucin attachment using DLS. As depicted in Fig. S17 (corresponding to below Fig. C) and Fig. S18 (corresponding to below Fig. D), increased particle size and a slight decrease of ~1-2 mV in zeta potential were found in comparison to native EcN. The conditions about how to determine the zeta potential using DLS were supplemented in the revised manuscript (highlighted in p.19) with below text: “The particle sizes and zeta potentials of thiolated bacteria were determined by dynamic light scattering (DLS) in double distilled H₂O (ddH₂O) at room temperature”.

We also considered the reviewer’s suggestion toward the use of chemical techniques for characterization. FTIR spectra were recorded to identify the presence of SH groups after thiolation. However, we cannot specifically identify the existence of SH at the predicted position (~2500-2600 cm⁻¹) after thiolation (below Fig. E). We would like to emphasize that different from substances with high purity, biosystems, such as living bacterial cells, are much more complex and contain various compositions and interferences, making it difficult to identify the existence of specific chemical groups and molecules by FTIR, XRD, NRM, etc. Actually, the tools that we applied, such as SH-specific clickable dyes and DTNB, are gold standards for SH quantification and more sensitive and selective for investigating SH and its reactions, especially in complex living systems.

3. Mucin are a highly structurally diverse family of glycoproteins and perhaps the structure depicted in Figure 1 and S1, and description on p.5 is too simplistic. Are the disulphide regions of the mucin accessible? Similar to the point above there could be steric hindrance/reaction by/with the sugar chains hindering disulphide formation.

How specific really is the thiolation reaction? Finally, what was the purity of the commercially used mucin? The reviewer's experience is that these can be contaminated with surfactants.

Response: We agree with the reviewer's point that the structure depicted in Fig. 1 and S1 are simplistic. The reason we simplified the structure was to solely highlight the reaction between mucin-associated disulfides and thiol groups on bacterial surface without causing potential confusing. According to the reviewer's suggestion, we have supplemented detailed description to clarify the structure of mucin in p.5 in the revised manuscript: "Mucin is an important category of large extracellular glycosylated proteins that are main organic components of mucus layer (36). The oligosaccharide chains consisting of 5-15 monomers exhibit moderate branching and are attached to the protein core by forming O-glycosidic bonds with the hydroxyl groups of serine and threonines and arranged in a "bottle brush" configuration (37). Mucin has cysteine-rich domains in N and C terminals that mediate chain extension by end-to-end disulfide linkage of mucin monomers (Fig. S1). Cysteine-rich regions are actively abundant and also reported as internal domains that contribute to disulfide side links between intermediate cysteine thiols (38)".

We also agree with the reviewer's point that some disulfides of mucin might be difficult to access due to steric hindrance. While, our experimental data well-demonstrated the occurrence of thiol-disulfide exchange reaction between EcN@SH and mucin (Fig. 4). First, thiolation level-dependent attachment proved that the interaction between thiolated bacteria and mucin could be tuned easily by varying the number of surface thiol groups (Fig. 4c and d). Second, to validate the formation of disulfide bonds between thiolated bacteria and mucin, L-ascorbic acid was added to the mixture prior to the reaction given that thiol-disulfide exchange could be dampened by reductive agents (Fig. 4g). At an L-ascorbic acid concentration of 0.3 mg/ml, thiolated EcN after reaction with Cy5.5 mucin presented a clear decrement in fluorescence intensity, which was further reduced with concentration increasing to 1 mg/ml (Fig. 4h and i).

In addition, we addressed the reviewer's concern about the specificity of the thiol reaction and have supplemented two extra experiments to validate the specific occurrence of thiol-disulfide exchange. First, dithiothreitol (DTT), which can reduce disulfide bonds, was applied to cleave the newly-formed disulfides between EcN@SH and mucin. As displayed in Fig. S19 (corresponding to below Fig. A), the attached mucin on EcN@SH could be totally cleaved by 0.1 mg/ml of DTT, verifying covalent bonding-mediated attachment of mucin on EcN@SH via thiol-disulfide exchange. Second, in addition to reductive agents, selenocystamine (SeCA), which contains diselenium and acts as a mimetic substrate competing with disulfides to exchange with thiols (Nature communications, 2021, 12, 163), was used to inhibit the reaction between EcN@SH and mucin. As displayed in Fig. S20 (corresponding to below Fig. B), 0.3 mg/ml of SeCA greatly blocked the conjugation of mucin on EcN@SH, further suggesting the occurrence of thiol-disulfide exchange reaction.

The commercial mucin was purchased from Sigma-Aldrich with high quality (M1778, Type III, bound sialic acid 0.5-1.5%, partially purified powder). We

acknowledged the reviewer’s point that the presence of surfactants in commercial mucin might be an issue. While, we would like to clarify that the purchased mucin was purified via ultrafiltration (100 kD, Millipore) to remove surfactants during the process of labeling. The associated description was highlighted in p.10 of the revised manuscript: “Cy5.5-labelled mucin (Cy5.5 mucin) was prepared by esterification with Cy5.5-NHS for 3 hours at 37 °C and the product was purified by ultrafiltration.”

4. A highlight of the paper is the preliminary in vivo data in mice that showed significant changes to jejunal physical properties, and slight reductions in inflammation as measured by reductions in expression of by one cytokine (TNF-alpha), when tested in a 5-fluorouracil-induced murine model of jejunal mucositis. It would have been good to increase the library of cytokine and chemokines tested to gauge more about the inflammatory pathways, and a discussion on why there was a small reduction in inflammation would be useful.

Response: Basing on the reviewer’s suggestion, the expression level of a major inflammatory cytokine, interleukin-6 (IL-6), in EcN@SH-dosed mice was determined

by ELISA, which showed an apparent reduction in contrast to those of PBS and EcN controls (Fig. S26, corresponding to below Fig. A). Additionally, the inflammation of jejunal tissue was assessed by myeloperoxidase (MPO) staining, showing significantly less MPO positive cells in jejunal lesion after EcN@SH treatment (Fig. S27, corresponding to below Fig. B and C).

Regarding the reduction in inflammation, together with the newly-supplemented data, our treatment outcome showed that EcN@SH displayed significant decrements in the levels of major inflammatory factors in comparison to EcN treatment, including 2.7-, 2.8-, and 3.5-fold reduction in TNF-alpha, IL-6, and MPO positive cells, with statistical p value less than 0.01, 0.05, and 0.01, respectively. These data supported the ability of EcN@SH to remit inflammation associated with 5-Fu-induced murine model of jejunal mucositis.

5. Grammar and spelling needs checking

Response: We have carefully checked and improved the English writing in the revised manuscript.

Overall this is quite a novel study but there are assumptions about the chemical modification on bacteria that need further clarification.

Response: We sincerely thank the reviewer again for taking her/his valuable time to review our work and providing useful comments that have substantially improved the quality of our manuscript. We hope that the reviewer could be satisfied with our major revision.

Reviewer #2 (Remarks to the Author):

In this manuscript, Luo et al. described a chemical reaction-mediated approach that facilitates covalent colonization of gut bacteria. The bacterial surface amine was converted into a free thiol in a one-step imidoester reaction. According to the authors, the free thiol group in the modified bacteria will help it spontaneously bond with a mucous layer rich in disulfides and remain attached longer. As a proof of concept, they applied this reaction to *E. coli* nissile (EcN). They demonstrated that thiolation of its surface increases its attachment time to mucus in the jejunum. Additionally, prolonged attachment ameliorates jejunal mucositis. There have been several studies showing that chemical modifications of bacterial surface structures either enhance or decrease their attachments to mucus (PMID: 11014524 and papers cited in the manuscript). There are several major issues that need to be addressed:

Response: We thank the reviewer very much for taking her/his valuable time to review our work and providing insightful comments and helpful suggestions on how to further refine the conclusions of our work. The raised concerns have been addressed as below, with detailed point-by-point responses.

1. The first comment is more conceptual. The definition of bacterial colonization is the ‘presence of a microorganism on/in a host, with growth and multiplication of the organism, but without interaction between host and organism’. Based on their description, the free thiol group is added in vitro, and only the modified bacteria are introduced in vivo. Each time the bacteria replicate in the intestine, the surface thiol groups will be diluted. The modification facilitates the attachment of the modified bacteria, but it is unclear whether this modification improves EcN colonization and facilitates the bacteria's growth and amplification to occupy an intestinal niche more efficiently. It would be more appropriate if the authors revised the manuscript to focus on the attachment/adhesion of EcN rather than colonization.

Response: We thank the reviewer for this important suggestion. We quite agree with this and have revised “colonization” to “attachment” or “localization” in the figures and associated texts throughout the manuscript.

2. The second comment is more technical. The chemical 2-iminothiolane they use is a small molecule. Did the authors examine whether this compound can be transported into EcN cells and interact with cellular free amines? This also applies to the Ellman's reagent they used to quantify the thiol numbers. The cells were treated with this reagent to determine the thiol number. 1M is a quite high concentration. Will the cells

get lysed during the treatment? There is a possibility that the total thiols quantified include both surface and intracellular free thiols.

Response: We are grateful to the reviewer for offering these insightful and valuable comments.

We have taken account of the reviewer's suggestion and conducted extra experiments using LC-MS to examine whether 2-iminothiolane could be transported into EcN cells and interact with cellular free amines. EcN cells were collected and washed with PBS for three times after incubation with 25 µg/ml of 2-iminothiolane for 90 minutes at room temperature. The obtained cells were further sonicated and freeze-thawed for three times to release intracellular substances. Potential 2-iminothiolane attached on cell membranes was removed by centrifugation at 15000 rpm for 10 minutes. The supernatant containing the released intracellular 2-iminothiolane was detected by LC-MS analysis. As shown in below Fig. A, the signal of 2-iminothiolane in the supernatant is extremely low (near undetectable), which suggested that 2-iminothiolane was difficult to transport into EcN cells when incubating with our optimized reaction conditions (25 µg/ml 2-iminothiolane for 90 minutes incubation at room temperature). As a positive control, when increasing 2-iminothiolane concentration to 100 µg/ml, we could detect intracellular signal of 2-iminothiolane using LC-MS. Two main reasons might account for limited internalization of 2-iminothiolane under the optimized thiolation conditions: 1) the presence of hindrance/resistance due to the membrane integrity of living bacteria, leading to less entry of 2-iminothilane; 2) the relatively low 2-iminothilane concentration of 25 µg/ml and short incubation time of 90 minutes, resulting in more possibility for 2-iminothiolane to interact with bacterial surface. Actually, the presence and location of thiols were visualized by LSCM imaging by applying an SH-selective and clickable dye, showing a uniform Cy5.5-marked layer on the surface of each thiolated bacterium, with negligible intracellular signal (Fig. 3c, g, k, o, s and Fig. S13, S14). These results were in consistence with LC-MS data, further verifying that the thiolation mainly occurred on bacterial surface.

Regarding the high concentration of 1 M, we apologize for the misleading of the working concentration of Ellman's reagent (DTNB). The final working concentration of DTNB to detect thiol number was 0.1 mM, not 1 M (stocking solution). The description was corrected in the section of "Determination of the number of thiol groups" in p.18 in the revised manuscript: "The bacteria were re-suspended in PBS with 3×10^8 CFUs/ml and DTNB was added to achieve a final working concentration of 0.1 mM, with the reaction proceeding for 2 hours at room temperature." We agree with the reviewer that there could be a possibility that the total thiols quantified by DTNB might include both surface and intracellular free thiols, especially when the cells were lysed by DTNB. To verify whether EcN cells were lysed during incubation with DTNB, the number of viable cells after treatment with 0.1 mM DTNB was examined. As displayed in Fig. S5 (corresponding to below Fig. B), viable cells remained near unchanged after treatment, suggesting that these cells were not lysed during DTNB treatment and the total thiols quantified mainly included surface free thiols.

3. The half-life of the modified bacteria in vitro and in vivo. The authors need to assess how long (or how many rounds of replication) it takes for the modified bacteria to lose their surface free thiol groups in vitro. In their in vivo experiments, the authors measured CFU levels at 1, 4, and 24 hours after gavage. As the CFU of modified EcN decreases, the authors need to determine how long it will take for EcN to return to its baseline level of attachment.

Response: We have taken into account the reviewer's suggestion and the in vitro change of surface free thiol groups of the modified bacteria over time in PBS was investigated using flow cytometry. As displayed in Fig. S15 (corresponding to below Figure), half of the newly-formed thiols disappeared with incubation time increasing to 2 hours. The levels of thiols decreased much more slowly after 2 hours incubation and returned to the baseline of native EcN with time prolonging to 12 hours. These results suggested relatively long reservation of the new-formed thiols on EcN@SH surface for thiol-disulfide exchange reaction.

Regarding how long it will take for EcN to return to its baseline level of attachment, we would like to emphasize that the attachment mediated by thiols could only maintain within 12 hours post oral administration, given the disappearance of newly-generated thiols observed in the in vitro experiment. Therefore, the increased number of EcN@SH after 12 hours was mainly ascribed to the early enhanced

reservation enabled by forming covalent bonds with mucin. According to our previous studies, oral delivered probiotics including EcN usually take about 5 days to return to the baseline level (Nature Communications 2019, 10, 5783; Science Advances 2020, 6, eabb1952).

4. It remains unclear how their method (covalent conjugation) is superior to prior approaches. The purpose of chemically modifying bacterial surfaces is to increase their attachment to epithelium. If the same result can be achieved via other methods (Page 4, lines 2 to 3), then the authors should state clearly and also demonstrate why their method is superior to previous ones. For example, why would noncovalent interactions between bacteria and epithelium result in inadequate interactions? While covalent bonding will not. All these chemical modifications are somewhat ‘temporary’ and do not occur in vivo. One round of bacterial replication may decrease surface free thiol groups by half. And in their experiments the modified bacterium was still administered daily.

Response: We thank the reviewer for drawing our attention to these issues. First, we would like to take this chance to emphasize that the current work describes a new concept of manipulating the interaction between bacteria and surroundings by spontaneously forming covalent bonds, which has been rarely reported before. Second, in terms of application, this approach can be applied to develop bacterial therapeutics, particularly with ability to increase localization on tissue surface with abundant mucosae. For instance, the obtained enhanced accumulation of probiotics in the jejunum after oral delivery has not been reported before. It is worth noting that the jejunum refers to a challenging location for transplanted microbiota to colonize due to the existence of unfavorable microenvironments and physical barriers, as reflected by its extremely low level of bacterial abundance. Lastly, the introduction of covalently linked thiol groups is more robust and durable compared to prior modifications, such as chemical coating and physical encapsulation, which result in noncovalent inadequate interactions and can be destroyed or eliminated after a few rounds of bacterial replication. Despite all these chemical modifications are relatively temporary

in contrast to synthetic bioengineering, potential safety issues of gene contamination associated with genetic engineering can be avoided. These have been highlighted in the sections of Introduction (p.4) and Conclusion (p.16) in the revised manuscript. Regarding the daily treatment, although the use of EcN@SH has not been optimized to reduce administration frequency, it did achieve a strikingly improved remission of jejunal mucositis in a murine model compared to unmodified EcN, which is a clinically used therapeutic agent. Given its significantly enhanced reservation in the intestine, there is still plenty room for optimizing the treatment regimen of these thiolated bacteria for further translation.

Again, we thank all the reviewers for taking their valuable time to review our manuscript. Their kind help and useful inputs are highly appreciated.

REVIEWERS' COMMENTS

Reviewer #1 (Remarks to the Author):

The authors have addressed all the concerns I raised, including supplementing the paper with additional experiments, so I am happy for the it to be accepted.

Reviewer #2 (Remarks to the Author):

The authors have nicely addressed all my comments. I recommend the manuscript be published as it is.